

# Analysis of local and systemic side effects of bacillus Calmette-Guérin immunotherapy in bladder cancer: a retrospective study in Türkiye

Ilkay Akbulut, İlker Ödemiş and Sabri Atalay

Department of Infectious Diseases and Clinical Microbiology, University of Health Sciences
Tepecik Training and Research Hospital, Izmir, Turkey

## ABSTRACT

**Background:** Bladder cancer is a growing health concern, especially in developing countries like Türkiye. Intravesical Bacillus Calmette-Guérin (BCG) immunotherapy is essential for reducing recurrence and progression in non-muscle invasive bladder cancer (NMIBC). However, it can cause local and systemic adverse effects linked to bacterial virulence, allergic reactions, or nosocomial infections. Data from randomized studies on BCG side effects are limited, with severe cases often reported in case studies. This study investigates the association between intravesical BCG immunotherapy and its adverse effects.

**Methods:** A retrospective analysis was conducted on 239 patients who underwent BCG immunotherapy between 2017 and 2024. Detailed demographic, clinical, and laboratory data were collected, and the adverse effects that developed following BCG therapy were evaluated. Descriptive statistics, including medians, counts, and percentage distributions, were calculated, and logistic regression analysis was performed to identify factors influencing the development of adverse effects.

**Results:** Adverse effects related to BCG immunotherapy were observed in 63.1% of the patients. The most common minor adverse effects were hematuria, dysuria, and cystitis, while major adverse effects included sepsis and lymphadenopathy. The analyses revealed that elevated aspartate aminotransferase (AST) levels and the presence of *Escherichia coli* and *Enterococcus faecalis* in urine cultures were significant risk factors for the development of adverse effects. Additionally, patients who underwent the 6th cycle of BCG therapy were found to have a higher risk of developing adverse effects compared to those who received fewer cycles.

**Conclusion:** BCG immunotherapy is an effective treatment method for NMIBC; however, the adverse effects that occur during treatment must be closely monitored. Elevated AST levels, the presence of specific pathogens in urine cultures, and the number of BCG doses administered are significant factors that increase the risk of adverse effects. These findings highlight the necessity for more careful monitoring throughout the treatment process.

Corresponding author
Ilkay Akbulut,
ilkayakbulutdr@gmail.com

## INTRODUCTION

Bladder cancer is among the most common urological malignancies worldwide, representing a significant public health concern. Globally, it ranks as the 10th most frequently diagnosed cancer, with an estimated 573,000 new cases and 213,000 deaths reported in 2020 (*Bray et al., 2024*). In developing countries like Türkiye, the incidence of bladder cancer has shown a steady increase, underscoring the critical need for early diagnosis and effective treatment strategies (*General Directorate of Public Health, 2018*). In cases of non-muscle invasive bladder cancer (NMIBC), where the risk of recurrence is high, the development of effective treatment strategies is of paramount importance.

Intravesical Bacillus Calmette-Guérin (BCG) immunotherapy is regarded as the gold standard in the treatment of NMIBC, particularly for intermediate-risk (IR) and high-risk (HR) patients, following surgical intervention (*EAU Guidelines, 2024*). Since its first use in the treatment of bladder cancer in 1976, BCG has played a crucial role in preventing the recurrence and progression of the disease in patients with high-risk NMIBC (*Redelman-Sidi, Glickman & Bochner, 2014*). This therapy is particularly effective in treating carcinoma *in situ* and recurrent high-risk tumors, significantly improving long-term disease-free survival rates (*Kamat et al., 2015*). The success of BCG immunotherapy largely depends on its ability to induce a local immune response within the bladder and to initiate a T cell-mediated anti-tumor response (*EAU Guidelines, 2024*).

However, BCG immunotherapy can lead to both local and systemic adverse effects during the treatment process, which can complicate the management of the therapy. Studies have shown that the incidence of side effects from BCG therapy varies significantly, and the continuation of therapy is often contingent upon the management of these side effects (*Zlotta, Fleshner & Jewett, 2009*). Adverse effects, particularly those related to excessive immune system activation, can reduce the quality of life for patients and negatively impact treatment adherence (*Green et al., 2019*).

The current European Association of Urology (EAU) guidelines recommend adapting BCG therapy based on patient risk stratification. IR-NMIBC patients are advised to receive 1 year of maintenance BCG therapy, while HR-NMIBC patients benefit from a 3-year maintenance regimen (*EAU Guidelines, 2024*). This stratified approach aims to balance treatment efficacy and tolerability, minimizing the risk of recurrence and progression.

This retrospective study evaluates the treatment outcomes of NMIBC patients who received BCG immunotherapy at Izmir Health Sciences University (HSU) Tepecik Training and Research Hospital between 2017 and 2024. The study provides a detailed analysis of the side effects encountered during BCG induction immunotherapy and the factors affecting these side effects. The findings aim to contribute to the existing literature on the feasibility and safety of BCG immunotherapy.

## MATERIALS AND METHODS

In this study, 239 patients diagnosed with NMIBC and treated with BCG immunotherapy at Izmir HSU Tepecik Training and Research Hospital between December 2017 and January 2024 were retrospectively reviewed through the hospital's information system. Written informed consent was obtained from the participants.

The EAU guidelines recommend stratified BCG schedules based on NMIBC risk levels. Patients with IR-NMIBC are advised to undergo 1-year maintenance BCG therapy, while patients with HR-NMIBC are recommended a 3-year maintenance regimen (*EAU Guidelines, 2024*). Regardless of histological differences, all patients in these risk groups undergo an initial induction therapy protocol. The induction therapy schedule follows the empirical 6-week regimen introduced by *EAU Guidelines (2024)*.

**BCG induction therapy:** BCG induction therapy involves intravesical instillations over a 6-week period to elicit a localized immune response and reduce recurrence risk.

**BCG maintenance therapy:** Maintenance therapy is applied based on risk stratification. Intermediate-risk patients typically receive a 1-year regimen, while high-risk patients are treated for up to 3 years, with periodic instillations at three-month intervals (*EAU Guidelines, 2024*).

Only side effects occurring during induction BCG immunotherapy were analyzed. Induction BCG instillations were given according to the standard 6-weekly schedule (*EAU Guidelines, 2024*).

The patients' demographic, clinical, and laboratory characteristics (aspartate aminotransferase (AST), alanine aminotransferase (ALT), creatinine, hemogram), existing comorbidities, side effects following BCG therapy (fever, cystitis, pneumonia, lymphadenopathy, hepatitis, sepsis), the number of BCG cycles administered, and the dosage at which side effects occurred were examined. Additionally, urine cultures were analyzed to identify the microorganisms present. The distribution of BCG immunotherapy administration over the years at our center was also reviewed, and changes in the treatment process were evaluated.

Adverse effects were classified as minor or major based on the literature. Minor side effects are those commonly observed after BCG therapy, typically requiring no treatment or manageable with minimal intervention. These side effects include hematuria, dysuria, fever, and cystitis (*EAU Guidelines, 2024*; *Herr & Donat, 2012*). Major side effects, on the other hand, are severe conditions requiring hospitalization or intensive medical intervention, such as sepsis, pneumonia, lymphadenopathy, hepatitis, and septic shock (*Pérez-Jacoiste Asín et al., 2014*). The reference ranges used were as follows: creatinine 0.7–1.2 mg/dL for males and 0.5–0.9 mg/dL for females; hemoglobin 13.2–17.3 g/dL for males and 11.8–16.0 g/dL for females; leukocyte count 4,000 to 10,000 cells/μL (*Pagana & Pagana, 2017*).

## Statistical analysis

This retrospective study utilized the Statistical Package for Social Sciences (SPSS) version 22 (IBM Inc., Armonk, NY, USA) for data analysis. Descriptive statistics were used to evaluate the data, which were presented as median (minimum-maximum values), count, and percentage distributions. The chi-square test was used for comparing categorical variables, while logistic regression analysis was employed to identify factors influencing the development of side effects.
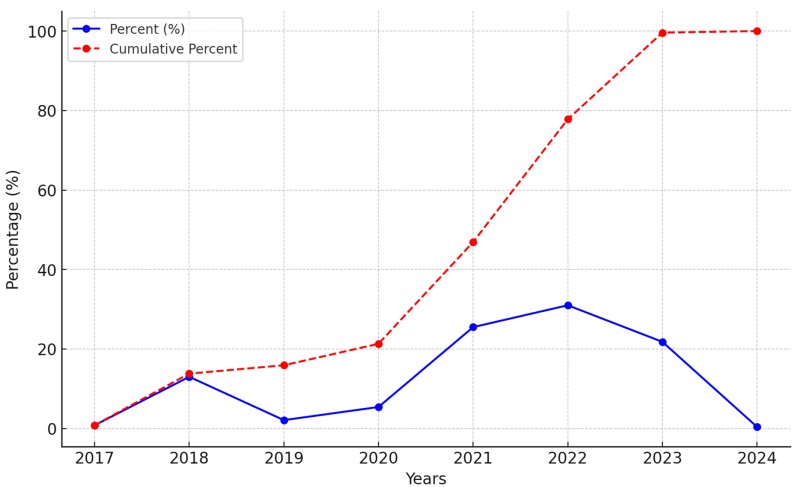

**Figure 1 BCG immunotherapy application rates by years.** The annual percentage of BCG immunotherapy applications (blue solid line) and their cumulative percentage (red dashed line) from 2017 to 2024.

**Ethical approval:** This study was approved by the Ethics Committee of the Health Sciences University Izmir Tepecik Training and Research Hospital on July 2, 2024, under decision number 2024/06–04.

## RESULTS

Data from 239 NMIBC-diagnosed patients who received intravesical BCG immunotherapy were retrospectively analyzed, focusing on their demographic characteristics, laboratory results, administered BCG cycles, and treatment-related side effects. The distribution by years is shown in Fig. 1. Of the patients included in the study, 86.6% ($n = 207$) were male, and 13.3% ($n = 32$) were female. The median age of the patients was 71 years, with an age range of 44 to 91 years. Regarding comorbidities, at least one comorbidity was identified in 42.6% ($n = 102$) of the patients. The most common comorbidities were hypertension (23.4%, $n = 56$), benign prostatic hyperplasia (22.1%, $n = 53$), and diabetes mellitus (10.4%, $n = 25$) (Table 1).

Laboratory findings showed that the median AST level was 22 U/L (range: 7–373), and the median ALT level was 21 U/L (range: 6–622). The median hemoglobin level was 12.6 g/dL (range: 6.0–16.9), with anemia present in 31.4% ($n = 75$) of the patients. The median leukocyte count was 10,000 cells/mm$^3$ (range: 3,100–23,100), with leukocytosis observed in 50.2% ($n = 120$) of the patients. The median creatinine level was 1.0 mg/dL (range: 0.1–2.2), and elevated creatinine levels were observed in 28.9% ($n = 69$) of the patients (Table 1).

Adverse effects related to BCG immunotherapy were observed in 63.1% ($n = 151$) of the patients. The most common minor side effects were hematuria (44.4%, $n = 106$), dysuria (28.0%, $n = 67$), cystitis (27.6%, $n = 66$), and fever (23.8%, $n = 57$). Major side effects were less common (3.3%, $n = 8$), with sepsis occurring in 3.7% ($n = 9$), lymphadenopathy in 3.3% ($n = 8$), hepatitis and septic shock each in 1.3% ($n = 3$). No cases of pneumonia or arthritis were reported (Table 1).

Table 1 Demographic, laboratory, and clinical characteristics of the patients.

| Variable | n (%) |
|---|---|
| Gender | |
| Male | 207 (86.6) |
| Female | 32 (13.3) |
| Age* | 71 (44–91) |
| Presence of comorbidities | 102 (42.6) |
| Types of comorbidities | |
| Diabetes mellitus (DM) | 25 (10.4) |
| Hypertension (HT) | 56 (23.4) |
| Benign prostatic hyperplasia (BPH) | 53 (22.1) |
| Laboratory values* | |
| Aspartate aminotransferase (AST, U/L) | 22 (7–373) |
| Alanine aminotransferase (ALT, U/L) | 21 (6–622) |
| Hemoglobin (Hgb, g/dL) | 12.6 (6.0–16.9) |
| White blood cell count (cells/mm$^3$) | 10,000 (3,100–23,100) |
| Creatinine (mg/dL) | 1.0 (0.1–2.2) |
| Presence of side effects | 151 (63.1) |
| Minor side effects | 151 (63.1) |
| Fever | 57 (23.8) |
| Cystitis | 66 (27.6) |
| Hematuria | 106 (44.4) |
| Dysuria | 67 (28.0) |
| Major side effects | 8 (3.3) |
| Sepsis | 9 (3.7) |
| Lymphadenopathy | 8 (3.3) |
| Hepatitis | 3 (1.3) |
| Septic shock | 3 (1.3) |
| Arthritis | 0 (0) |
| Pneumonia | 0 (0) |
| Positive urine culture | 72 (30.2) |
| Completion of 6 BCG therapies | 217 (90.8) |
| Association of side effects with BCG therapy | |
| 1st therapy | 18 (7.5) |
| 2nd therapy | 2 (0.8) |
| 3rd therapy | 18 (7.5) |
| 4th therapy | 23 (9.6) |
| 5th therapy | 3 (1.3) |
| 6th therapy | 12 (5.0) |

Note:
* Median (IQR) (min-max).

Urine culture results showed bacterial growth in 30.2% (*n* = 72) of the patients. The most frequently isolated microorganisms were *Escherichia coli* (10.9%, *n* = 26), *Enterococcus faecalis* (9.2%, *n* = 22), and Klebsiella species (4.6%, *n* = 11). Less commonly

**Table 2 Bacterial species identified in urine culture.**

| Bacterial species | n (%) |
|---|---|
| E. coli | 26 (10.9) |
| E. faecalis | 22 (9.2) |
| Klebsiella | 11 (4.6) |
| E. faecium | 3 (1.3) |
| E. cloacae | 3 (1.3) |
| M. morganii | 2 (0.8) |
| S. epidermidis | 2 (0.8) |
| S. agalactiae | 2 (0.8) |
| Acinetobacter spp | 1 (0.4) |

**Table 3 Multivariate analysis of variables affecting side effects.**

| Variables | n (%) | Univariate | | Multivariate | |
|---|---|---|---|---|---|
| | | OR (95% CI) | $p^*$ | OR (95% CI) | $p^*$ |
| AST | | – | 0.03 | 0.964 [0.934–0.994] | 0.019 |
| E. coli identification | | | | | |
| Yes | 21 | 0.373 [0.135–1.027] | 0.001 | 0.329 [0.114–0.947] | 0.039 |
| No | 5 | Reference | | | |
| E. faecalis identification | | | | | |
| Yes | 1 | 0.071 [0.009–0.539] | 0.001 | 0.060 [0.008–0.462] | 0.007 |
| No | 21 | Reference | | | |
| BCG 4th therapy | | | | | |
| Yes | 68 | 0.619 [0.559–0.685] | 0.001 | – | 0.999 |
| No | 8 | Reference | | | |
| BCG 6th therapy | | | | | |
| Yes | 55 | 14.054 [1.856–106.407] | 0.001 | 8.622 [1.061–70.086] | 0.044 |
| No | 21 | Reference | | | |

Notes:
95% CI, 95% confidence interval; OR, odds ratio; AST, aspartate aminotransferase.
* $p$ value of ≤0.05 (statistically significant).

isolated organisms included *Enterococcus faecium*, *Enterobacter cloacae*, *Morganella morganii*, *Staphylococcus epidermidis*, *Streptococcus agalactiae*, and Acinetobacter species (Table 2).

The completion rate of BCG therapy cycles was high, with 90.8% ($n = 217$) of the patients successfully completing the six-cycle treatment regimen. However, the risk of developing side effects increased with the number of cycles administered. Notably, a significant increase in side effects was observed after the 4th cycle, with 9.6% ($n = 23$) of patients experiencing side effects during the 4th cycle and 5% ($n = 12$) during the 6th cycle.

Multivariable analysis of factors influencing the development of side effects revealed that elevated AST levels significantly increased the risk of side effects (OR: 0.964; 95% CI [0.934–0.994]; $p = 0.019$). Additionally, patients with *E. coli* growth in their urine cultures

had a significantly higher risk of developing side effects (OR: 0.329; 95% CI [0.114–0.947]; $p = 0.039$). Similarly, the presence of *E. faecalis* in urine cultures was associated with an increased risk of side effects (OR: 0.060; 95% CI [0.008–0.462]; $p = 0.007$). Patients who completed the 6th cycle of BCG therapy were found to have a significantly higher risk of developing side effects compared to those who received fewer cycles (OR: 8.622; 95% CI [1.061–70.086]; $p = 0.044$) (Table 3).

## DISCUSSION

In this retrospective study, multivariate analysis results indicated that elevated AST levels and the presence of *Escherichia coli* and *Enterococcus faecalis* in urine cultures were significant risk factors for the development of side effects in the treatment of NMIBC. Furthermore, the risk of adverse effects was notably higher in patients who completed the 6th cycle of BCG therapy compared to those who received fewer cycles. These findings align with the existing literature, while also offering new perspectives on certain critical points. BCG immunotherapy has long been a standard treatment for high-risk superficial bladder cancers and carcinoma *in situ*, primarily due to its ability to induce a T-cell-mediated immune response within the bladder, thereby reducing the risk of disease recurrence (*Redelman-Sidi, Glickman & Bochner, 2014*; *Poletajew, Zapała & Radziszewski, 2017*; *Yücetaş & Toktaş, 2011*). However, the local and systemic side effects of BCG, particularly severe ones, can limit the feasibility of this treatment (*Fuge et al., 2015*; *Pérez-Jacoiste Asín et al., 2014*).

Our study found that 63.1% of patients developed side effects following BCG immunotherapy, with the majority being minor, although major side effects were also observed. The literature reports incidences of BCG-associated cystitis ranging from 57% to 91%, hematuria from 26% to 55%, and fever from 28% to 73% (*Poletajew, Zapała & Radziszewski, 2017*; *Han & Pan, 2006*). Similarly, our study frequently observed local side effects such as hematuria (44.4%) and dysuria (28%). The frequency of local side effects like cystitis and hematuria is related to the local inflammation induced by BCG within the bladder and is generally considered a marker of the immune response (*Poletajew, Zapała & Radziszewski, 2017*; *Han & Pan, 2006*). On the other hand, serious systemic side effects are relatively rare but can be life-threatening for patients. For instance, in our study, serious side effects such as sepsis (3.7%) and lymphadenopathy (3.3%) were identified. The literature also indicates that systemic BCG side effects are rare but can be severe when they do occur, underscoring the importance of close monitoring during and after BCG immunotherapy for the early detection and management of side effects (*Pérez-Jacoiste Asín et al., 2014*).

Our study identified certain biochemical and microbiological factors that increase the risk of side effects following BCG therapy. Elevated AST levels were found to significantly increase the risk of side effects (*Fujita et al., 2021*). This finding suggests that liver function may play an important role in the development of side effects during BCG therapy. Elevated AST levels typically indicate liver damage or an inflammatory response in the liver. The immunomodulatory effects of BCG therapy may trigger an inflammatory response in the liver, thereby increasing the risk of side effects. The impact of BCG

immunotherapy on the liver has been discussed in the literature, with attention drawn to the risk of hepatotoxicity (*Stamatakos et al., 2024*). This finding suggests that BCG therapy should be closely monitored in patients with impaired liver function, and AST levels may serve as a potential marker during treatment.

The presence of *E. coli* (OR: 0.329; 95% CI [0.114–0.947]; $p$ = 0.039) and *E. faecalis* (OR: 0.060; 95% CI [0.008–0.462]; $p$ = 0.007) in urine cultures significantly increases the risk of side effects. These findings suggest that microbial colonization during BCG therapy may play a critical role in the development of side effects (*Pérez-Jacoiste Asín et al., 2014*). Microorganisms such as *E. coli* and *E. faecalis* may enhance inflammatory responses on the bladder epithelium, potentially adversely affecting BCG's immunological effects (*Stamatakos et al., 2024*). The literature also reports that urinary tract infections (UTIs) can increase the risk of side effects during BCG therapy (*Oddens et al., 2013*). UTIs may modulate the immune response within the bladder, potentially exacerbating inflammation and leading to more severe side effects (*Han & Pan, 2006*). Therefore, effective management of UTIs before and during BCG therapy, including appropriate antibiotic treatment, is of great importance (*Stamatakos et al., 2024*).

The risk of side effects was found to be significantly higher in patients who completed the 6th cycle of BCG therapy compared to those who received fewer cycles. This finding suggests that the cumulative effects of BCG therapy may increase the risk of side effects. The literature indicates that the immune-enhancing effects of BCG may accumulate over the course of treatment, leading to more intense inflammatory reactions in the bladder wall, which can exacerbate local side effects and even result in systemic side effects. The observed increase in side effects after the 4th cycle highlights the need to carefully evaluate the duration of treatment (*Poletajew, Zapała & Radziszewski, 2017*). In this context, further research is needed to develop shorter but effective treatment protocols that minimize the risk of side effects (*Oddens et al., 2013*).

In our study, 69 (28.9%) patients with elevated creatinine levels, 75 (31.4%) anemic patients, and 120 (50.2%) patients with leukocytosis were identified. This suggests that BCG therapy may adversely affect renal function and hematological parameters. The existing literature supports these findings and underscores the importance of careful laboratory monitoring during BCG therapy (*Stamatakos et al., 2024*). When assessing urine culture results, bacterial growth was observed in 30.2% of cases. This finding aligns with the literature on urinary tract infections observed during BCG immunotherapy, with pathogens such as *Escherichia coli* and *Enterococcus faecalis* frequently isolated (*Stamatakos et al., 2024*). However, the literature also indicates that BCG therapy efficacy is not negatively impacted in patients with positive urine cultures; in fact, some studies report lower recurrence rates due to a more intense immune response (*Redelman-Sidi, Glickman & Bochner, 2014*; *Lamm et al., 2000*). Nevertheless, as seen in our study, the presence of *E. coli* and *E. faecalis* in urine cultures is among the factors that increase the risk of side effects. As far as we know, *Escherichia coli* and *Enterococcus faecalis* are considered to be common pathogens of urinary tract infections. However, studies have shown that uropathogenic *E. coli* infection promotes a paracellular permeability defect associated with the failure of umbrella cell tight junction formation and umbrella cell

sloughing (*Wood et al., 2012*). Furthermore, suppression of NF-κB activation by UPEC leads to increased type-1 fimbria-mediated apoptosis of urothelial cells and decreased inflammatory cytokine production and neutrophil recruitment (*Billips et al., 2007*). In terms of *Enterococcus faecalis*; the metalloprotease GelE, produced by commensal strains of *E faecalis*, contributes to development of chronic inflammation in mice that are susceptible to inflammation by impairing epithelial barrier integrity (*Steck et al., 2011*). All these studies support the finding that these two pathogens increase the risk of side effect development in BCG immunotherapy. This finding highlights the importance of carefully managing potential infections during BCG therapy (*Decaestecker & Oosterlinck, 2015*).

The detrimental effects of antibiotic administration on infective complications, particularly in high-risk patients, can complicate treatment outcomes. In this context, inappropriate use of antibiotics may enhance bacterial colonization and adversely impact the immunological effects of BCG therapy (*Liedberg, Xylinas & Gontero, 2024*). Additionally, broad-spectrum antibiotics have been reported to suppress immune responses, exacerbating both local and systemic inflammation (*Aubert et al., 2024*). Therefore, careful evaluation of antibiotic use during BCG therapy and avoidance of potential drug interactions are critical. Identifying infection risk factors prior to therapy and ensuring appropriate antibiotic management can enhance treatment efficacy and prevent adverse effects. Moreover, further studies are warranted to better understand the potential effects of microorganisms on immune responses. In this regard, optimizing antibiotic use not only for infection management but also for limiting side effects during BCG therapy is essential.

Emerging molecular targets and novel therapeutic alternatives offer promising solutions for patients unfit for BCG instillations. Recent advances in understanding bladder cancer pathophysiology have identified potential molecular targets such as FGFRs, immune checkpoint inhibitors, and other signaling pathways, which are being evaluated in clinical trials (*Claps et al., 2023*). For instance, targeted therapies that inhibit FGFR pathways or utilize immune modulators such as anti-PD-1 and anti-PD-L1 antibodies have shown significant potential in improving outcomes for patients unsuitable for standard BCG therapy (*Hurle et al., 2020*). Moreover, alternative strategies, including the use of nanoparticles for drug delivery and mucoadhesive systems, are being explored to enhance the precision and efficacy of bladder cancer treatment (*Tang et al., 2021*). These advancements underline the importance of integrating molecular insights into therapeutic decisions, particularly for patient populations with limited tolerance for conventional therapies like BCG.

## LIMITATIONS

This study has several limitations. First, its retrospective design, relying on data retrieved from past records, increases the potential for biases such as missing data. Additionally, the study was conducted in a single institution, which may limit its ability to reflect regional variations in treatment practices and patient demographics. While the sample size is representative of the hospital population, the generalizability of the findings to a broader

population is limited. Furthermore, the study did not account for potential confounding factors such as treatment adherence levels or other health conditions that could influence the development of side effects. Therefore, future prospective studies involving larger and more diverse patient populations with longer follow-up periods are necessary. Such studies would provide more comprehensive insights into the risks and benefits of BCG immunotherapy in the treatment of non-muscle invasive bladder cancer.

## CONCLUSION

This study reaffirms that BCG immunotherapy is an effective treatment for non-muscle invasive bladder cancer but highlights the importance of vigilance regarding potential side effects during therapy. Close monitoring of patients before and after treatment, careful observation of laboratory values, and prompt intervention when side effects occur are essential to improving treatment outcomes. Future research should focus on developing more specific guidelines for the prevention and management of these side effects.

## ABBREVIATIONS

| | |
|---|---|
| **AST** | Aspartate aminotransferase |
| **ALT** | Alanine aminotransferase |
| **BCG** | Bacillus Calmette-Guérin |
| **NMIBC** | Non-muscle invasive bladder cancer |
| **HSU** | Health Sciences University |

### Funding

The authors received no funding for this work.

### Competing Interests

The authors declare that they have no competing interests.

### Author Contributions

- Ilkay Akbulut conceived and designed the experiments, performed the experiments, prepared figures and/or tables, and approved the final draft.
- İlker Ödemiş performed the experiments, analyzed the data, prepared figures and/or tables, and approved the final draft.
- Sabri Atalay analyzed the data, prepared figures and/or tables, authored or reviewed drafts of the article, and approved the final draft.

### Human Ethics

The following information was supplied relating to ethical approvals (*i.e.*, approving body and any reference numbers):

Ethics Committee of the Health Sciences University Izmir Tepecik Training and Research Hospital (2024/06–04).

## Data Availability

The raw data is available in the Supplemental Files.

## Supplemental Information

Supplemental information for this article can be found online at http://dx.doi.org/10.7717/peerj.18870#supplemental-information.

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
