# Peer review of "Analysis of local and systemic side effects of bacillus Calmette-Guérin immunotherapy in bladder cancer: a retrospective study in Türkiye"

_PeerJ, doi:10.7717/peerj.18870_

## Round 0.1 · original submission · Major Revisions

Please address the comments of both reviewers. As noted by Reviewer 1, the study must be updated to current guidelines

Reviewer 1 ·

Basic reporting

Please pay attention to the following comments for revision.

Authors should adapt BCG schemes to the current SOC of NMIBC both considering intermediate risk (IR) and high risk (HR) NMIBC. Authors should refer to current EAU Guidelines: 1-year maintenance (IR-NIMBC) vs. 3-year maintenance (HR-NMIBC).

Authors should report the cumulative number of instillations as potential risk of infective complications. Moreover, to give a more homogeneous overview compared to the current literature authors should introduce the concept of both "induction" and "maintenance" regimen. It seems that authors evaluated only the first 6 instillations (induction phase) that now are defined as "non adequate exposure".

I suggest a further paragraph in the discussion section considering the potential detrimental effect - on prognosis - of antibiotic administration in such infective complications. Authors should refer to DOI: 10.3389/fonc.2023.1240378; DOI: 10.1016/j.euf.2023.11.007. Main point of discussion is to identify pts who are more at risk of such a grueling course and to avoid drug interferences that could impair treatment outcomes.

Authors should update their references considering novel emerging molecular targets and alternatives to such a group of pts who are potentially unfit for BCG instillations. Consider the novel insights as DOI: 10.3390/ijms241612596; DOI: 10.1002/bco2.28.

Experimental design

Observational study that could be more refined by adhering to the current Guidelines (EAU).

Validity of the findings

Should be adapted to the current Guidelines.

Additional comments

None.

Reviewer 2 ·

Basic reporting

.

Experimental design

.

Validity of the findings

.

Additional comments

This manuscript (title: Analysis of Local and Systemic Side Effects of Bacillus Calmette-Guérin Immunotherapy in Bladder Cancer: 7 Years Clinical Experience) aimed to investigate the relationship between intravesical BCG immunotherapy and its associated adverse effects.
Several critical issues warrant further attention:

1.The Background section in the Abstract is too long; please shorten it.
2.In the Introduction, please cite references when presenting epidemiological data (e.g., PMID: 38572751). Authors should mention that BCG treatment should be applied following surgery (e.g., PMID: 38819629).

3.Why did the authors not reference the EAU classification for adverse effects?
The dose and frequency of BCG should be clearly specified.
4.What is the duration of the observed adverse effects? Induction period? Induction + maintenance period? How long was the maintenance period?
5.The authors should present histological features of the included patients, such as WHO grade, histological type, and T stage. NMIBC with histological variants shows different response rates to BCG.
6.Escherichia coli and Enterococcus faecalis are commonly considered primary pathogens in urinary tract infections. The authors regard them as risk factors. What distinguishes them from urinary infections? Please provide a full explanation.

Therefore, I recommend that this article should undergo major revisions before publication.

---

## Round 0.2 · accepted · Accept

After revisions, all reviewers agreed to publish the manuscript. I also reviewed the manuscript and found no obvious risks to publication. Therefore, I also approved the publication of this manuscript.

Reviewer 1 ·

Basic reporting

Authors provided a revised version of the manuscript. No further comments.

Experimental design

Valid

Validity of the findings

Valid

Additional comments

None.

Reviewer 2 ·

Basic reporting

This manuscript has improved and can be accepted.

Experimental design

no comment

Validity of the findings

no comment

Additional comments

no comment